

# Naringenin attenuates inflammation and apoptosis of osteoarthritic chondrocytes via the TLR4/TRAF6/NF-κB pathway

Yan Wang[1,*], Zhengzhao Li[2,*], Bo Wang[3], Ke Li[3] and Jiaxuan Zheng[4]

[1] Department of Hand Surgery, Hainan General Hospital (Hainan Affiliated Hospital of Hainan Medical University), Haikou, China
[2] Department of Emergency Surgery, Hainan General Hospital (Hainan Affiliated Hospital of Hainan Medical University), Haikou, China
[3] Department of Sports Medicine, Hainan General Hospital (Hainan Affiliated Hospital of Hainan Medical University), Haikou, China
[4] Department of Pathology, Hainan General Hospital (Hainan Affiliated Hospital of Hainan Medical University), Haikou, China
* These authors contributed equally to this work.

Corresponding author
Jiaxuan Zheng, zjxhhhyh@163.com

## ABSTRACT

Naringenin is a flavonoid extracted from the seed coat of *Anacardiaceae* plants. Increasing evidence indicates that it has several properties of biological significance, such as anti-infection, sterilization, anti-allergy, antioxidant free radical, and anti-tumor. However, its effect on osteoarthritis has not been elucidated properly. In this study, the treatment of primary chondrocytes with interleukin (IL)-1β was found to increase the secretions of IL-6, tumor necrosis factor (TNF)-α, and cyclooxygenase-2 (COX-2). Further, the mRNA expression of matrix metalloproteinase ((MMP)3, MMP9, and MMP13), the protein expression of Recombinant A Disintegrin And Metalloproteinase With Thrombospondin 5 (ADAMTS5), and cell apoptosis increased; the protein expression of Collagen II decreased. The injury of primary chondrocytes induced by IL-1β was reversed under the intervention of naringenin; this reversal was dose-dependent. The mechanistic study showed that naringenin inhibited the toll-like receptor 4 (TLR4)/TNF receptor-associated factor 6 (TRAF6)/NF-κB pathway in IL-1β-stimulated primary cells, and LPS, a TLR4 activator, reversed this inhibitory effect. In addition, a mouse model of osteoarthritis was established and treated with naringenin. The results revealed that naringenin alleviated the pathological symptoms of osteoarthritis in mice, reduced the expression of TLR4 and TRAF6, and the phosphorylation of NF-κB in knee cartilage tissue. It also inhibited the secretion of inflammatory factors, reduced extracellular matrix degradation, and decreased the protein expression of cleaved caspase3.
In conclusion, the findings of this study suggest that naringenin may be a potential option for the treatment of osteoarthritis.

## INTRODUCTION

Osteoarthritis is a chronic joint disease characterized by the degeneration of articular cartilage (*Tim et al., 2022*). Its prevalence increases rapidly with age. According to the World Health Organization, osteoarthritis ranks fourth in female chronic diseases and eighth in male chronic diseases. In China, the incidence of knee osteoarthritis is as high as 49% in the population older than 60 years (*Quicke et al., 2022*; *Allen, Thoma & Golightly, 2022*).

Currently, the pathogenesis of osteoarthritis is inconclusive. It is believed that an imbalance of extracellular matrix metabolism, chondrocyte apoptosis, and autoimmune disorders may cause osteoarthritis. Various pathogenic factors, such as cytokines, inflammatory transmitters, immune factors, and active proteases, can trigger these imbalances. Chondrocyte apoptosis is also considered to play a key role in osteoarthritis development.

Flavonoids are naturally occurring compounds found in many edible plants. They possess important biological properties, such as anti-infection, sterilization, anti-allergic, antioxidant free radicals, and anti-tumor. Increasing evidence indicates that flavonoids help in the recovery of injured osteoarthritis chondrocytes. *Chu et al. (2020)* reported that casticin-treated IL-1β-stimulated ADTC5 cells displayed decreased levels of reactive oxygen species and secretion of proinflammatory cytokines. They also reported that casticin inhibited oxidative stress and reduced inflammation in a mouse model of osteoarthritis, suggesting that casticin alleviates osteoarthritis-related cartilage degradation by inhibiting the ROS-mediated NF-κB signaling pathway (*Chu et al., 2020*). Another study reported that verlot extract rich in flavonoids inhibited cyclooxygenase synthesis and significantly improved motor function and allodynia in a rat model of sodium monoiodoacetate-induced osteoarthritis (*Vasconcelos et al., 2019*). Further, icariin inhibited the progression of osteoarthritis by inhibiting pyroptosis mediated by NLRP3/caspase 1 signaling *in vitro* and *in vivo* osteoarthritis models (*Zu et al., 2019*).

Naringenin is an aglycone obtained from naringenin by hydrolyzing a molecule of rhamnose and glucose. It is a monomeric flavonoid with a molecular formula of $C_{15}H_{12}O_5$ and a relative molecular weight of 273.25. Naringenin has been reported to have anti-inflammatory properties. A previous study showed that naringenin treatment significantly reduced the secretion levels of IL-6 and TNF-α in TGF-β stimulated fibrotic NRK-52E cells. The mechanism suggested this was mediated through the transforming growth factor (TGF-β)/Smad pathway (*Wang et al., 2021*). *Zhao et al. (2020)* reported that naringenin restored endothelial barrier integrity by downregulating proinflammatory factors in oxidized low-density lipoprotein-treated human umbilical vein endothelial cells, suggesting that naringenin may have a therapeutic effect on endothelial injury-related diseases. *Xu, Zhang & Sun (2017)* showed that naringenin could exert anti-inflammatory effects by reducing the production of prostaglandin E2, NO, IL-6, and TNF-α in LPS-treated Raw 264.6 cells. Another study reported that oral naringenin reduced cartilage matrix degradation in mice and delayed the progression of osteoarthritis. The protective effect of naringenin on cartilage and chondrocytes is probably due to the inhibition of

NF-κB pathway (*Zhao et al., 2016*). As a type of flavonoid, naringenin has received extensive attention from researchers for its anti-inflammatory ability. In hepatocytes, naringenin has been shown to reduce the production of pro-inflammatory cytokines, such as TNF-α, IL-6, and IL-1β, by inhibiting the NF-κB signaling pathway (*Chtourou et al., 2015*). In a mouse model of collagen-induced arthritis, the inhibitory effect of naringenin on LPS-induced JNK/MAPK and p65/NF-κB signaling prevents dendritic cell maturation and reduces the production of proinflammatory cytokines (*Li et al., 2015*). In summary, the anti-inflammatory mechanism of naringenin is related to its regulation of inflammatory factors such as TNF-α, IL-1β, IL-6, and p65/NF-κB signaling pathways. Since osteoarthritis is closely associated with inflammation, it was speculated that naringenin would play an essential anti-inflammatory role in inhibiting osteoarthritis progression.

In this study, an osteoarthritis cell model was prepared by treating primary chondrocytes with IL-1β; the effects of different naringenin concentrations on the secretion of inflammatory factors, matrix degradation, and apoptosis of chondrocytes were determined. Further, the anti-inflammatory effect of naringenin on osteoarthritis *in vivo* by establishing a mouse model of osteoarthritis. The results of this study demonstrate that naringenin has excellent potential for osteoarthritis treatment.

## MATERIALS AND METHODS

### Cell culture

The knee joint cartilage tissues of healthy male C57BL/6 mice (6 weeks; weight 18–20 g) were collected. The following procedure was performed: The skin and muscle around the knee joint were separated. Next, the hyaline cartilage was separated and put into a Petri dish containing PBS. Subsequently, the soft tissue around the hyaline cartilage was isolated, placed in a sterile EP tube, and sheared. The cartilage tissue was resuspended with 1 ml of 0.25% trypsin and digested. Digestion was terminated after 30 min, and the trypsin was discarded after centrifugation at 300 g for 10 min in a 4 °C centrifuge. Next, 3 ml of 1.5% collagenase type II was added, resuspended, transferred to a centrifuge tube, and placed in a cell incubator for digestion for 6 h. The centrifuge tube was centrifuged at 300 g for 10 min at 4 °C in a low-speed centrifuge. The supernatant was discarded, and the cells were resuspended using Dulbecco's Modification of Eagle's Medium (DMEM). The cell suspension was transferred to a bottle for culture. When the cells proliferated to 90% confluence, they were digested with trypsin and passaged. A high glucose medium containing 10% Fetal bovine serum DMEM was used to culture primary chondrocytes. In addition, a double antibiotic solution consisting of 100 U/ml of penicillin-streptomycin was added to the culture medium (15140122; Gibco, Billings, MT, USA) and incubated in a 37 °C constant temperature incubator containing 5% carbon dioxide ($CO_2$). When the cells were passaged to P3–P5, they were used for subsequent studies.

### Animals

The male C57BL/6 mice (6 weeks, 18–20 g; Henan Experimental Animal Center, Henan, China) were randomly divided into three groups ($N = 8$ per group) including sham group

(the medial meniscotibial ligament was visualized but not transected), osteoarthritis group (the model was constructed by the destabilization of the medial meniscus method) and naringenin group (naringenin 10 mg/(kg•d) was administered by gavage once a day). The animal model of osteoarthritis was established through meniscus destabilization surgery. The specific modeling procedure is as follows: All animals were weighed and skinned before surgery, and 10% chloral hydrate solution was injected intraperitoneally into anesthetized rats at 0.35–0.45 mL/kg. Subsequently, a longitudinal incision was made about 1 cm medial to the right kneecap, and layered dissection was conducted until the knee joint cavity was exposed. In the sham surgery group, the tissues were sutured layer by layer without any tissue damage. In the model group, the patella was laterally dislocated, and the knee joint was flexed to fully expose the anterior cruciate ligament and medial meniscus. Ophthalmic scissors were then used to remove the medial meniscus. Finally, the patella was repositioned, and the joint cavity was sealed with physiological saline. The incision was stitched layer by layer. After the mice woke up, it was put back in the cage to move freely. Penicillin injections were administered to prevent infection for the first 3 days after surgery, and the filler was replaced every 3 days. Mice in the osteoarthritis model group were given the same amount of normal saline daily, while mice in the sham group were not given any treatment. Naringenin treatment was administered on the second day after modeling, once a day for 14 days. The mice were euthanized, and the knee cartilage tissues were collected for subsequent experiments. The mice were housed in a clean and well-ventilated animal environment at 20 ± 2 °C, with a relative humidity of 60–70% and a day/night cycle of 12/12 h. They had free access to water and food. Eight weeks after surgery, the mice were euthanized to harvest knee cartilage tissues. The cervical dislocation method was used to euthanize the mice in the experimental group. All experiments were conducted in accordance with the Animal Ethics Committee of Hainan General Hospital.

## RT-qPCR

A total of 1 ml of Trizol (15596-018; Invitrogen, Waltham, MA, USA) was added per will in six-well cell culture plates to extract total cellular RNA. The extracted RNA was reverse transcribed into cDNA using primescript RT kits (RR037A; Takara, Kusatsu, Japan) according to the manufacturer's instructions. The reverse transcription procedure was as follows: A mixture containing 2 μl of 5 × PrimeScriptBuffer, 0.5 μl of PrimeScriptRT Enzyme Mix 1, 0.5 μl of Oligo Dt Primer (50 μM), 0.5 μl of Random 6 mers (100 μM), and 50 ng of RNA. The mixture was supplemented to 10 μl with RNA-free $dH_2O$, and the reaction conditions were set to 37 °C, with 3 cycles of 15 min for reverse transcription reaction. Next, the prepared PCR reaction solution was placed in a Real-time PCR instrument for the PCR amplification reaction. The reaction conditions were the following: Pre denaturation at 95 °C for 4 min; 95 °C denaturation for 30 s, 57 °C annealing for 30 s, 72 °C extension for 30 s, a total of 40 cycles. The relative expression levels were normalized by using the $2^{-\Delta\Delta Ct}$ method.

## Western blotting

The protein was extracted with 1 ml of RIPA lysis buffer containing 10 μL of protease inhibitor (R0010; Solarbio, Beijing, China). The primary antibodies were as follows: GAPDH (1:3,000), Collagen II (1:1,500), ADAMTS5 (1:200), cleaved caspase3 (1:500), TRAF6 (1:3,000), NF-κB (1:1,500) and NF-κB (phospho S536, 1:1,000). The membranes were incubated with secondary antibody (1:2,500) for 2 h. The membrane was removed from the secondary antibody solution, placed again in the TBST solution, and shaken for 10 min on a shaking table—this was repeated thrice. A developing solution was prepared with a 1:1 volume ratio of liquid A and liquid B of the chemiluminescent liquid and placed on ice to avoid light. Next, the prepared chemiluminescence solution was evenly dropped onto the surface of the PVDF film. The film was exposed in the chemiluminescence imaging system instrument, and the program was set to collect three pictures per second and expose them for 5 min. The bands were saved, and the data was uploaded. The Image J Lab software to analyze the band grayscale value. Using the GAPDH protein as the internal reference, the relative expression ratio of the target protein was calculated.

## CCK-8 assay

Routine digestion and resuspension were done to prepare single-cell suspension. The cell density was adjusted to $2 \times 10^4$/ml. A total of 100 μl of cell suspension was added to each well in a 96-well plate. Three duplicate wells were set up. The 96-well plate was placed in a cell incubator for routine cultivation. The culture medium was changed once a day. According to the established time points in the experiment, 10 μl of CCK-8 solution was added to each well after 24, 48, and 72 h of cultivation, and the culture continued for 4 h. After incubation, 100 μl of solution was added to each well and continued to incubate for 4 h. The absorbance was measured at a wavelength of 450 nm. Repeat each experiment three times and take the average value.

## Flow cytometry

The fluorescein FITC labeled Annexin V is a phospholipid-binding protein with a high affinity for phosphatidylserine. It binds to the cell membrane of early apoptotic cells through phosphatidylserine exposed outside the cell. It is a sensitive indicator for detecting early apoptosis. The cell culture medium was retrieved into flow cytometry tubes and washed twice with PBS; the PBS was recycled into the corresponding flow cytometry tubes. The cells were digested with trypsin without EDTA and incubated. The digestion was terminated with serum. The digested cells were collected, and cell suspension was added to the corresponding flow cytometry tubes centrifuged at 1,000 rpm for 6 min at 4 °C. The supernatant was discarded, and 2 ml of PBS was added to resuspend the cells. The cells were centrifuged at 800 rpm for 5 min at 4 °C—This process was repeated thrice. Next, 195 μl Annexin V-FITC binding solution was added to resuspend the cells gently, then 5 μl Annexin V-FITC was added, and the cells were incubated at room temperature in the dark for 10 min. Subsequently, the cells were centrifuged at 200 g for 5 min, and the supernatant was discarded. Next, 190 μl Annexin V-FITC binding solution was added to resuspend the

cells, 10 µl propidium iodide staining solution was added, and cell apoptosis was detected by flow cytometry.

## Histopathologic assessment

Gradient dewaxing was performed on the slices using ethanol. The paraffin sections were dewaxed according to the following procedure: The paraffin sections were put into xylene I for 10 min, xylene II for 10 min, xylene III for 10 min, absolute ethanol I for 5 min, absolute ethanol II for 5 min, 90% alcohol for 5 min, 80% alcohol for 5 min, 70% alcohol for 5 min, 50% alcohol for 5 min. The liquid on the slide was then gently shaken dry. Hematoxylin was added dropwise, and after staining for 3 min, the color development was stopped with tap water. It was then differentiated using 1% hydrochloric acid alcohol and 1% ammonia water to reverse blue. Next, the tissues were stained with eosin for approximately 1 min. After dehydrating, the slices were sealed with adhesive and placed in a well-ventilated area to dry. Finally, the samples were imaged with a microscope.

## Statistical analysis

All statistical analyses were performed using the SPSS software (ver. 22.0; SPSS, Chicago, IL, USA). The quantitative data from three independent experiments were expressed as mean ± standard deviation (mean ± SD). The Shapiro-Wilk test was used to verify whether the data was normally distributed. Levene's test was used to verify the homogeneity of variances. The parameter test was used for data that conformed to the normal distribution. The comparison between two groups was performed by Student $t$-test, and the comparison between multiple groups was performed by one-way analysis of variance (ANOVA), followed by an LSD test for *post hoc* analysis. A nonparametric test was used for data that did not conform to normal distribution. The Mann Whitney U test was used to compare two groups, while the Kruskal Wallis test was used to compare multiple groups. $p < 0.05$ was considered statistically significant.

# RESULTS

## Naringenin enhances cell viability

Figure 1A shows the chemical structure of naringenin ((4′,5,7-Trihydroxyflavanone, $C_{15}H_{12}O_5$)). The cytotoxicity of naringenin on primary chondrocytes at concentrations of 10, 20, 30, 40, and 50 µM was determined. The results revealed that naringenin was toxic to the cells at 40 and 50 µM concentrations (Fig. 1B). However, cell viability was significantly enhanced with naringenin in a dose-dependent manner at concentrations of 10, 20, and 30 µM (Fig. 1C).

## Naringenin attenuates inflammation response

IL-1β-treated primary chondrocytes were incubated with 10, 20, and 30 µM naringenin. With increasing naringenin dose, the secretion of IL-6 (Fig. 2A), TNF-α (Fig. 2B), and COX-2 (Fig. 2C) decreased significantly in a dose-dependent manner.

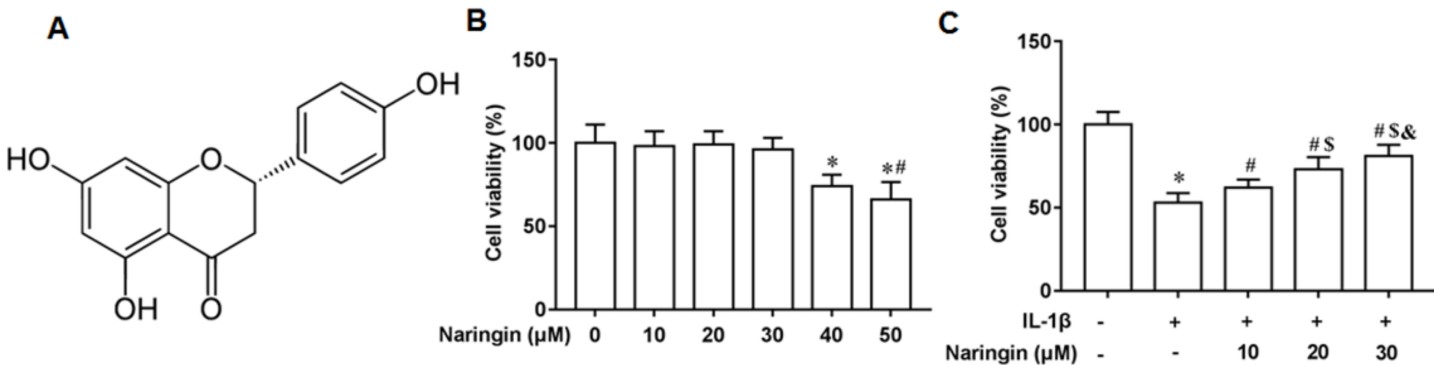

**Figure 1 Effect of naringenin on the viability of IL-1β-treated primary chondrocytes.** (A) The chemical structure (4′,5,7-Trihydroxyflavanone, $C_{15}H_{12}O_5$) of naringenin. (B) Cell toxic effect. (C) Cell viability. $N = 6$. *$p < 0.01$ compared with Control group. #$p < 0.01$ compared with IL-1β group. $p < 0.01$ compared with IL-1β +10 μM Naringin group. &$p < 0.01$ compared with IL-1β +20 μM Naringin group.

**Figure 2 Effect of naringenin on the inflammatory response of IL-1β-treated primary chondrocytes.** The primary chondrocytes were treated with IL-1β alone or together with naringin (10, 20 and 30 μM). (A–C) The IL-6, TNF-α and COX-2 secretion levels. $N = 6$. *$p < 0.01$ compared with Control group. #$p < 0.01$ compared with IL-1β group. $p < 0.01$ compared with IL-1β +10 μM Naringin group. &$p < 0.01$ compared with IL-1β +20 μM Naringin group.

## Naringenin alleviates cell matrix degradation

The IL-1β-treated primary chondrocytes were treated with 10, 20, and 30 μM naringenin. The mRNA expression levels of matrix metalloproteinases, including MMP3 (Fig. 3A), MMP9 (Fig. 3B), and MMP13 (Fig. 3C), decreased after naringenin treatment and significantly correlated with the dosage. Moreover, naringenin treatment significantly reversed the decrease of Collagen II protein expression (Figs. 3D and 3E) and increased ADAMTS5 protein expression (Figs. 3D and 3F) induced by IL-1β stimulation.

## Naringenin alleviates cell apoptosis

Treating primary IL-1β-treated chondrocytes with 10, 20, and 30 μM of naringenin revealed that apoptosis decreased with increasing naringenin concentration (Fig. 4A).

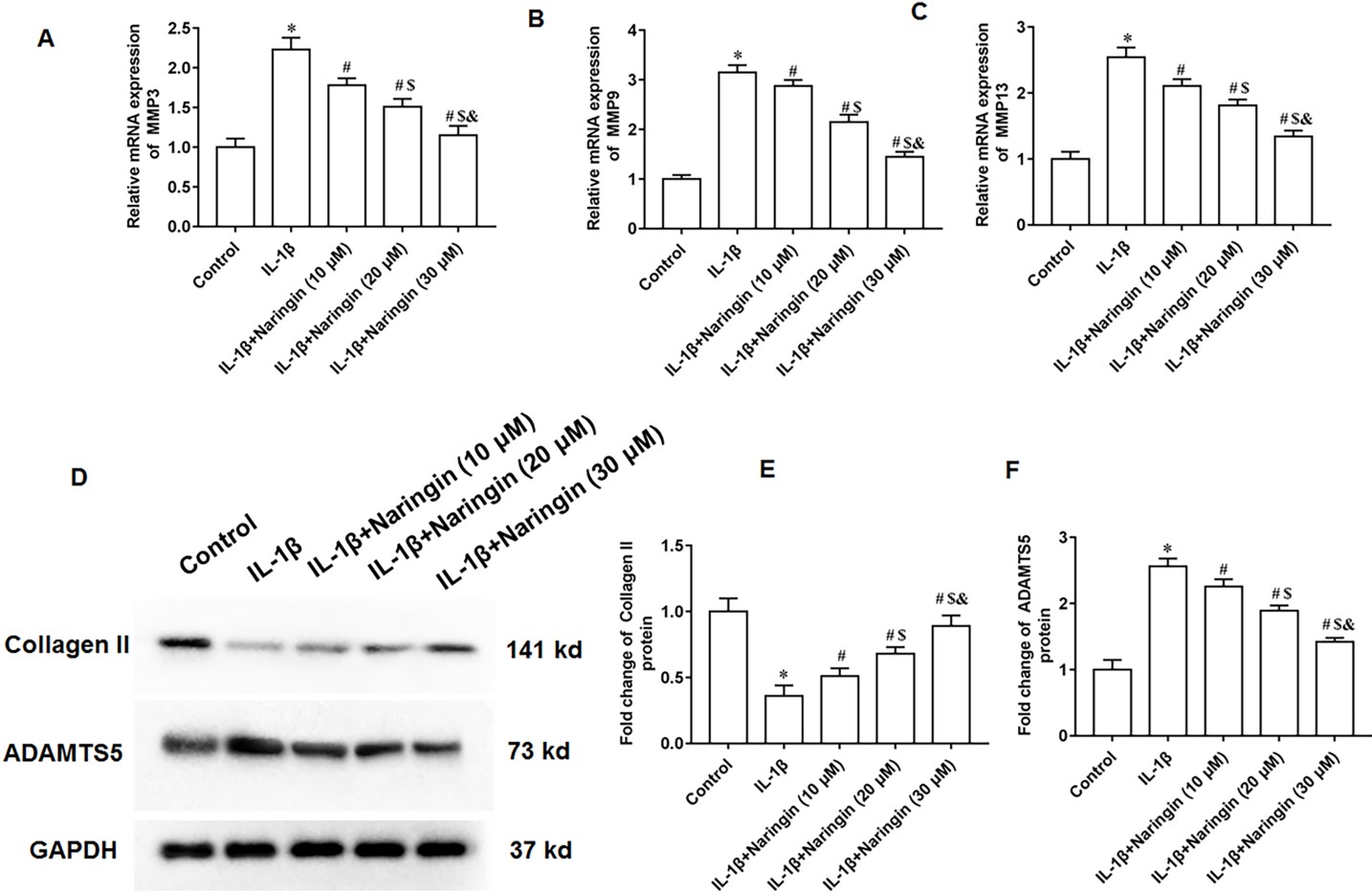

**Figure 3 Effect of naringenin on matrix degradation in IL-1β-treated primary chondrocytes.** The primary chondrocytes were treated with IL-1β alone or together with naringin (10, 20 and 30 μM). (A–C) The protein expression of MMP3, MMP9 and MMP13 was detected by Western blotting. (D–F) The Collagen II and ADAMTS5 protein expression. $N = 6$. [*]$p < 0.01$ compared with Control group. [#]$p < 0.01$ compared with IL-1β group. [$]$p < 0.01$ compared with IL-1β +10 μM Naringin group. [&]$p < 0.01$ compared with IL-1β +20 μM Naringin group.

Moreover, IL-1β treatment promoted the expression of cleaved caspase3 in cells, and naringenin treatment significantly reversed the effect of IL-1β stimulation (Fig. 4B).

## Naringenin attenuates cell injury *via* TLR4/TRAF6/NF-κB pathway

IL-1β-treated primary chondrocytes were treated with naringenin alone or simultaneously with TLR4 activators (LPS). IL-1β treatment promoted TLR4 and TRAF6 protein expression and upregulated the phosphorylation level of NF-κB. Although naringenin treatment reversed the effect of IL-1β, its effect was neutralized after LPS intervention (Figs. 5A and 5B). Further, naringenin treatment enhanced the viability (Fig. 5C), reduced IL-6 (Fig. 5D) and COX-2 (Fig. 5E) secretion, decreased the content of MMP13 (Fig. 5F), promoted the secretion of Collagen II (Fig. 5G) and inhibited cell apoptosis (Fig. 5H) in IL-1β-treated primary chondrocytes. However, LPS neutralized these effects of naringenin.

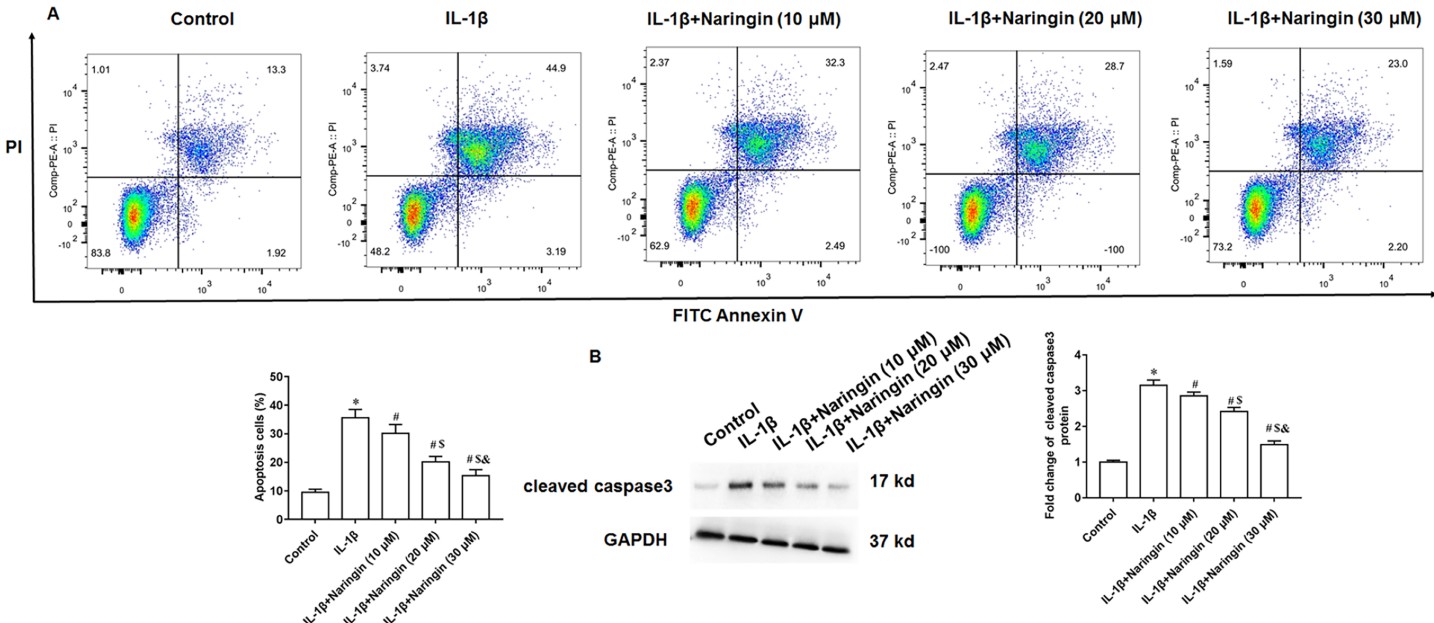

**Figure 4 Effect of naringenin on apoptosis in IL-1β-treated primary chondrocytes.** The primary chondrocytes were treated with IL-1β alone or together with naringin (10, 20 and 30 µM). (A) Cell apoptosis. (B) The protein expression of cleaved caspase3 was detected by Western blotting. $N = 6$. *$p < 0.01$ compared with Control group. #$p < 0.01$ compared with IL-1β group. $$p < 0.01$ compared with IL-1β +10 µM Naringin group. &$p < 0.01$ compared with IL-1β +20 µM Naringin group.

## Naringenin alleviates osteoarthritis in mice

Naringenin treatment ameliorated the damage to knee cartilage tissues in mice. Figure 6A shows the representative images of hematoxylin-eosin staining of knee cartilage tissues. In osteoarthritis mice, Naringenin treatment significantly reduced TLR4 and TRAF6 protein expression and NF-κB phosphorylation (Fig. 6B). ELISA results indicated that naringenin inhibited IL-6, COX-2, and MMP13 secretion and promoted Collagen II secretion (Figs. 6C and 6D). Using the OARSI scoring system to evaluate the pathological grades of the knee joints in the osteoarthritis mice, it was found that naringenin treatment improved the symptoms of osteoarthritis mice (Fig. 6E). Naringenin treatment also inhibited cleaved caspase3 expression in the cartilage tissues of mice with osteoarthritis (Figs. 6B and 6F).

## DISCUSSION

The inflammatory factor IL-1β is reported to be an important factor in initiating osteoarthritis. It increases iNOS and COX-2 in Osteoarthritis Chondrocytes, promoting the secretion of NO and PGE2 by chondrocytes. Collagen II and aggrecan are the main components of the cartilage extracellular matrix. NO promotes cartilage extracellular matrix degradation by inhibiting the synthesis of collagen II and aggrecan; PGE2 also contributes to the degradation of cartilage extracellular matrix. IL-1β also stimulates the gene expression and protein secretion of other proinflammatory and chemokines, including TNF-α and IL-6. At present, IL-1β is usually used to induce osteoarthritis models in chondrocytes. Thus, inhibiting IL-1β secretion and IL-1β-induced degradation

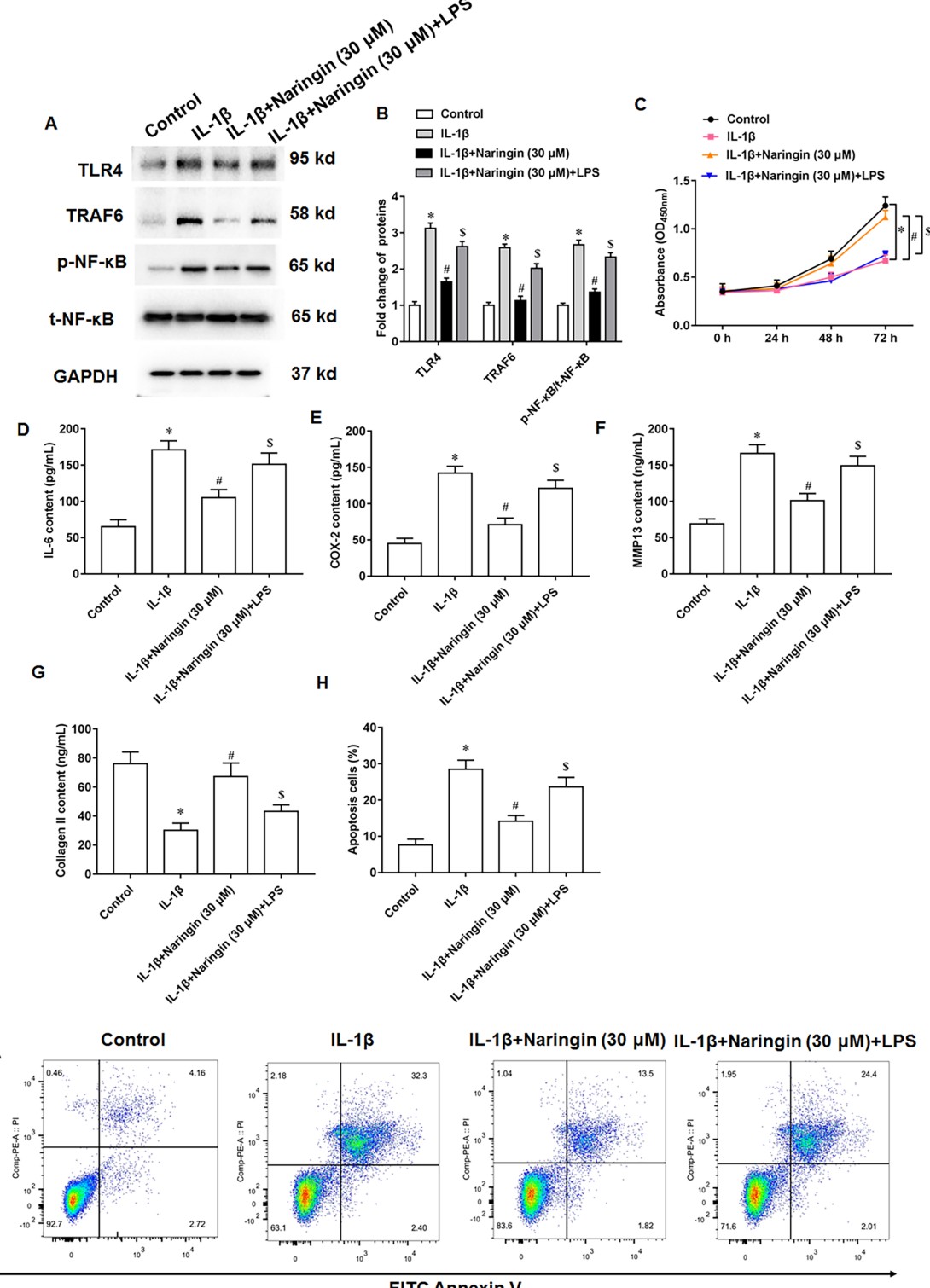

**Figure 5 Naringenin regulates IL-1β-treated primary chondrocytes *via* TLR4 signaling pathway.** The IL-1β-treated primary chondrocytes were treated with naringenin alone or simultaneously intervened with TLR4 activators (LPS). (A and B) The protein expression of TLR4, TRAF6 and total NF-κB, and the phosphorylation level of NF-κB was detected by Western blotting. (C) Cell viability. (D–G) The secretion levels of IL-6, COX-2, MMP13 and Collagen II. (H) Cell apoptosis. $N = 6$. $^*p < 0.01$ compared with Control group. $^\#p < 0.01$ compared with IL-1β group. $^\$p < 0.01$ compared with IL-1β + 30 μM Naringin group.

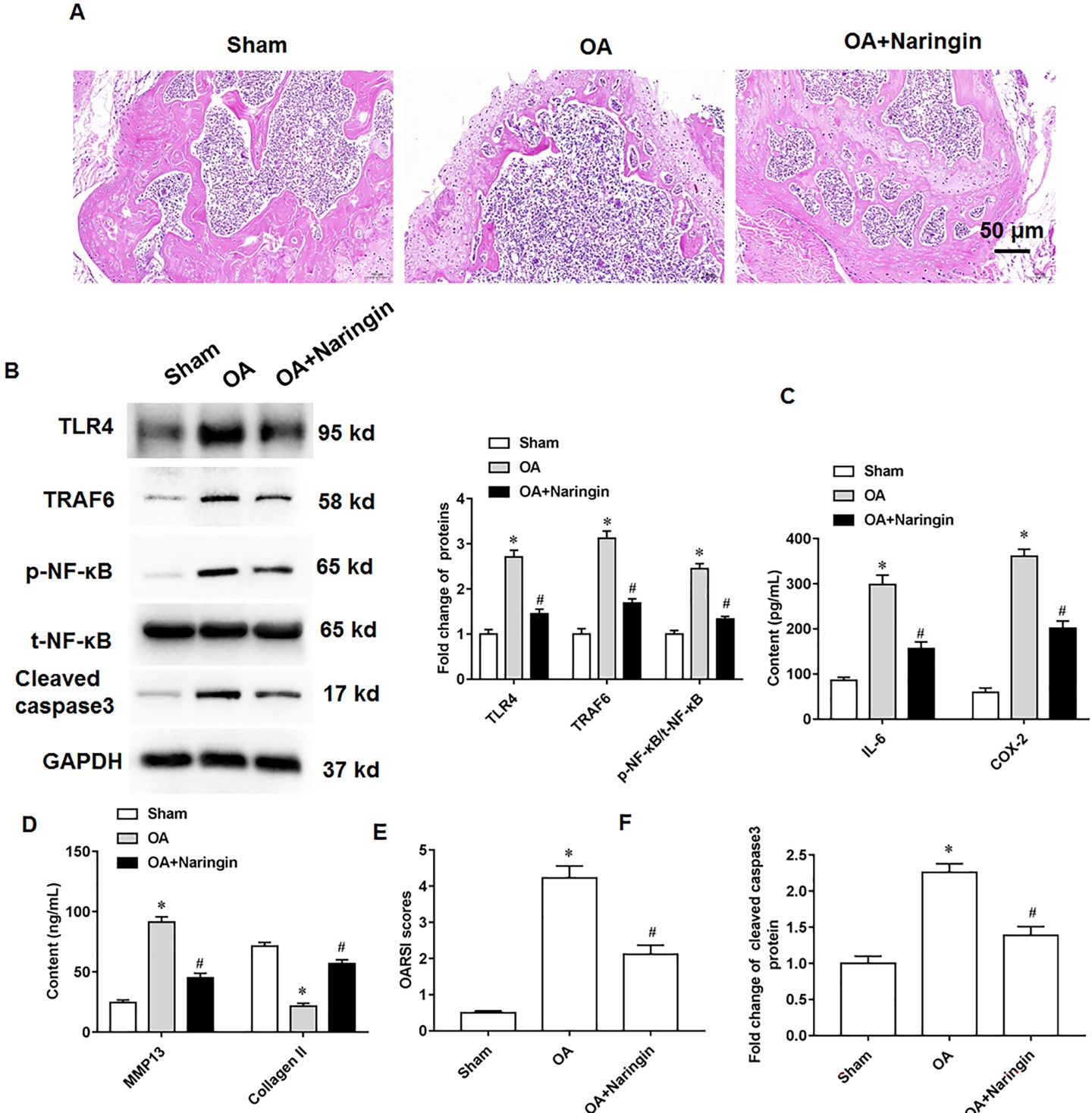

**Figure 6 Therapeutic effects of naringenin on osteoarthritis in mice.** Sham group (the medial meniscotibial ligament was visualized but not transected), osteoarthritis group (the model was constructed by the destabilization of the medial meniscus method) and naringenin group (treated with 10 mg/kg naringin by intraperitoneal injection). (A) Representative images of HE staining of mouse knee joint cartilage tissues. (B) The protein expression of TLR4, TRAF6, total NF-κB and cleaved caspase3, and the phosphorylation level of NF-κB was detected by western blotting. (C and D) IL-6, COX-2, MMP13 and Collagen II secretion levels. (E) OARSI scores of mice. (B and F) The cleaved caspase3 expression. $N = 8$. $^*p < 0.01$ compared with Sham group. $^#p < 0.01$ compared with OA group.

of the cartilage matrix may provide an effective therapeutic target for the prevention and treatment of osteoarthritis.

Chondrocytes play a key role in the synthesis and turnover of extracellular matrix in cartilage tissue. As the only cell morphology in normal articular cartilage, they also maintain the integrity of the extracellular matrix. It also plays a key role in maintaining the integrity of cartilage structure and bearing the weight of cartilage (*Li et al., 2021*; *Gu et al., 2021*). Studies have shown that chondrocyte apoptosis is one of the main factors in bone and joint osteoarthritis. Compared to ordinary cell apoptosis, cartilage cell apoptosis in osteoarthritis possesses some unique features: (1) Both chondrocyte and apoptotic bodies possess alkaline phosphatase and trinucleotide phosphate dehydrogenase activities, which can induce calcium deposition; (2) in the event of cartilage cell apoptosis, the apoptotic body cannot be carried away by the macrophages and is left inside the joint cartilage, affecting the normal physiological function of the joint cartilage. The apoptotic body is released into the joint space and eliminated only when the cartilage matrix is degraded. Apoptosis in articular cartilage is closely correlated with matrix degradation. When articular cartilage cells over apoptosis, matrix production decreases, gradually creating a vicious cycle. Multiple *in vitro* and *in vivo* studies indicate that the proinflammatory cytokine IL-1β mediates the destruction of articular cartilage and promotes chondrocyte apoptosis (*Chen et al., 2021*). Additionally, TNF-α cannot directly cause extracellular matrix degradation in osteoarthritis chondrocytes but rather induces chondrocyte production of MMP3 through TNF-α receptor P55, leading to degradation of the cartilage matrix and subsequent chondrocyte apoptosis. Therefore, the apoptosis of chondrocytes in the progression of osteoarthritis does not occur immediately, which leads to a slow process of osteoarthritis cartilage degradation (*Weber, Bolia & Trasolini, 2021*).

IL-1β induces the production of NO and prostaglandin E2 by upregulating the expression of inducible nitric oxide synthase and cyclooxygenase 2. NO and prostaglandin E2 promote cell decomposition, which can induce chondrocyte apoptosis by stimulating the ROS and mitogen protein kinase pathways (*Zhou et al., 2020*; *Bai et al., 2020*). Moreover, IL-1β synergizes with other cytokines in the osteoarthritis progression, resulting in the metabolic imbalance of chondrocytes. Additionally, the high expression of matrix metalloproteinases (MMPs) further damages the integrity of the cartilage extracellular matrix and the internal stability of cartilage tissue by accelerating the breakdown of proteoglycans and type II collagen in the cartilage matrix (*Li et al., 2020*; *Dai et al., 2019*). Our results are consistent with previous results. In this study, we found that matrix metalloproteinases MMP3, MMP9, and MMP13 increased, while collagen II and adamts5 decreased in IL-1β-stimulated primary chondrocytes, suggesting that IL-1β accelerated matrix degradation in chondrocytes.

TLRs are expressed in articular cartilage and synovial fibroblasts of patients with osteoarthritis; TLR4 is the primary receptor form of TLRs in chondrocytes. A study showed that linalool inhibited the LPS-induced overproduction of NO, prostaglandin E2, IL-6, and TNF-α in chondrocytes; the mechanism study showed that linalool blocked the activation of NF-κB by inhibiting the formation of TLR4/myeloid differentiation protein-2 dimer complex, thus, delaying osteoarthritis progression (*Qi et al., 2021*). *Zhang et al.*

*(2020)* found that the expression of ARFRP1 and TLR4 was increased, while that of miR-15a-5p was decreased in osteoarthritis cartilage tissue. ARFRP1 silencing alleviated chondrocyte injury, and mechanistic studies showed that ARFRP1 induced chondrocyte injury by regulating TLR4/NF-κB axis (*Zhang et al., 2020*). TLR4 and myeloid differentiation factor 88 (MyD88) dependent pathways mediated apoptosis and inflammatory activation are critical signal transduction systems involved in the progression of osteoarthritis. In the TLR4-induced MyD88-dependent signaling pathway, the downstream TRAF6 is located at the intersection of the TLR4/MyD88 signaling pathway. A study reported that avicularin could inhibit extracellular matrix degradation and inflammatory response by blocking the TRAF6/MAPK pathway (*Zou et al., 2021*). *Jiang et al. (2021)* found that TRAF6 silencing inhibited the production of MMP-13 and IL-6 induced by LPS and reduced cell apoptosis. They also reported that IκBα degradation and p65 nuclear transport were also inhibited, indicating that TRAF6 silencing inhibited the activation of the NF-κB pathway by LPS (*Jiang et al., 2021*). Consistent with the studies of other scholars, we found that in the *in vitro* cell model of osteoarthritis induced by IL-1β, TLR4 signal was activated, TRAF6 protein expression was increased, the phosphorylation level of NF-κB was increased, and the inflammation, apoptosis and matrix degradation of chondrocytes were aggravated, and under the action of naringenin, TLR4 pathway was inhibited, and the inflammation, apoptosis and matrix degradation of chondrocytes induced by IL-1β were also alleviated.

## CONCLUSIONS

In this study, we found that naringenin enhanced the viability of IL-1β-treated primary chondrocytes and alleviated the inflammatory response, matrix degradation and apoptosis, which was mediated by inhibiting the TLR4/TRAF6/NF-κB pathway. The *in vivo* results showed that naringenin treatment significantly improved osteoarthritis in mice. Considering the anti-inflammatory effect of the flavonoid naringenin in osteoarthritis progression, it was speculated that naringenin may also have anti-inflammatory and therapeutic effects on other orthopedic inflammatory diseases, including intervertebral disc degeneration and rheumatoid osteoarthritis. Other flavonoids with a chemical structure similar to naringenin may also play an essential role in inflammatory orthopedic diseases- This will be investigated further in future studies. The findings of this study will provide new ideas for the treatment of osteoarthritis.

### Funding

This study was supported by the Hainan Provincial Natural Science Foundation of China (821MS127). The funders had no role in study design, data collection and analysis, decision to publish, or preparation of the manuscript.

## Grant Disclosures

The following grant information was disclosed by the authors:
Hainan Provincial Natural Science Foundation of China: 821MS127.

## Competing Interests

The authors declare that they have no competing interests.

## Author Contributions

- Yan Wang conceived and designed the experiments, performed the experiments, authored or reviewed drafts of the article, and approved the final draft.
- Zhengzhao Li performed the experiments, analyzed the data, prepared figures and/or tables, and approved the final draft.
- Bo Wang conceived and designed the experiments, prepared figures and/or tables, authored or reviewed drafts of the article, and approved the final draft.
- Ke Li conceived and designed the experiments, analyzed the data, prepared figures and/or tables, and approved the final draft.
- Jiaxuan Zheng performed the experiments, analyzed the data, authored or reviewed drafts of the article, and approved the final draft.

## Animal Ethics

The following information was supplied relating to ethical approvals (*i.e.*, approving body and any reference numbers):

All samples obtained in this study were approved by the ethics committee of the Hainan General Hospital.

## Data Availability

The raw data is available in the Supplemental Files.

## Supplemental Information

Supplemental information for this article can be found online at http://dx.doi.org/10.7717/peerj.16307#supplemental-information.

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
