# Peer review of "Naringenin attenuates inflammation and apoptosis of osteoarthritic chondrocytes via the TLR4/TRAF6/NF-κB pathway"

_PeerJ, doi:10.7717/peerj.16307_

## Round 0.1 · original submission · Minor Revisions

Please respond and make appropriate revisions based on the Reviewers' suggestions and my comments (below). This will greatly improve the quality of the manuscript.

My comments:

This study not only uncovers a novel use for naringenin in treating osteoarthritis, expanding its known biological functions, but also seems to delve into the specific pathways that naringenin affects, adding to our understanding of its mechanism of action in osteoarthritis treatment.

Issues to be revised or improved:

1. The statement "the prevalence of osteoarthritis ranks fourth in women and eighth in men" is unclear without specifying what it is ranked against (e.g., all diseases, chronic conditions, etc.). This should be clarified for better understanding.

2. Line 46: "Mainstream scholars believe that the imbalance of extracellular matrix metabolism, chondrocyte apoptosis and autoimmune disorders caused by a variety of pathogenic factors, including various cytokines, inflammatory transmitters, immune factors and active proteases." This sentence seems to be missing a verb, creating confusion about the relationship between these factors. Should be revised to [Mainstream scholars believe that osteoarthritis may be caused by an imbalance of extracellular matrix metabolism, chondrocyte apoptosis, and autoimmune disorders. These imbalances can be triggered by a variety of pathogenic factors, including various cytokines, inflammatory transmitters, immune factors, and active proteases].

3. While the introduction references previous studies on osteoarthritis, it may be helpful to provide more in-depth explanations of what was found in those studies and how they relate to the current research.

4. The phrase "We intervened IL-1β-treated primary chondrocytes with 10, 20, 30 μM naringenin, respectively," is repeated several times. Consider rephrasing or summarizing common procedures to avoid repetition.

5. The phrase "We intervened IL-1β-treated primary chondrocytes with 10, 20, 30 μM naringenin, respectively," is repeated several times. Consider rephrasing or summarizing common procedures to avoid repetition.

6. The Discussion section should tie back directly to the specific results of this study. It would be beneficial to integrate the findings shown here with previous research more seamlessly and to highlight what the current study adds to the existing body of knowledge.

7. The Conclusions section succinctly summarizes the findings, but it might benefit from a more detailed reflection on the broader implications and potential applications of the study.

8. [More and more evidences show] should be [More and more evidence].

9. [autoimmune disorders caused] should be [autoimmune disorders is caused].

10. [increasing evidence have] should be [increasing evidence has].

11. [free accesses to] should be [free access to].

12. [add 100 μl of cell suspension to each well in a 96 well plate, and set up] should be [adding 100 μl … plate, and setting up].

13. [PBS to resuspent] should be [PBS to resuspend].

14. Line 141: [Cell] should be [Cells].

15. Line 141, 144, 148, 155: [10, 20, 30 μM naringenin]: Add [and] before 30.

16. [damage of knee cartilage] should be [damage to knee cartilage].

17. [expression of ARFRP1 and TLR4 were] should be [expression of ARFRP1 and TLR4 was].

18. [miR-15a-5p were] should be [miR-15a-5p was].

**Language Note:** PeerJ staff have identified that the English language needs to be improved. When you prepare your next revision, please either (i) have a colleague who is proficient in English and familiar with the subject matter review your manuscript, or (ii) contact a professional editing service to review your manuscript. PeerJ can provide language editing services - you can contact us at copyediting@peerj.com for pricing (be sure to provide your manuscript number and title). – PeerJ Staff

Reviewer 1 ·

Basic reporting

The manuscript is written in clear English and the articulation of complex concepts and technical details is well done. The comprehensive introduction draws on the background knowledge of primary chondrocytes, their functions and the detrimental influences of the proinflammatory cytokine IL-1β. However, the manuscript would benefit from a more structured representation of its context within the broader field of osteoarthritis research. Regarding the identification of the knowledge gap, the authors focus on the significance of TLR4/TRAF6/NF-κB pathway and its role in mediating inflammatory responses and apoptosis. The study further solidifies understanding in this area by exploring the effects of naringenin, marking a significant contribution to the field. The technical standards employed and the experimental design are sound, with a systematic evaluation of naringenin's effects in both in vitro and in vivo settings. The study's methods are articulated with a high level of detail supporting replication of the study. The underlying data appears to be robust and well-controlled, although further details on any statistical analyses performed would strengthen the study. Overall, the findings contribute valuable insights into the plausible use of naringenin for osteoarthritis treatment.

Experimental design

1. Specify the source of the articular linelicartilage tissue.
2. Specify the concentration of Fetal bovine serum (FBS) in the DMEM.
3. Specify the volumes and concentrations of the RIPA lysis buffer.
4. Specify the type and concentration of ethanol used for gradient dewaxing.

Validity of the findings

1. Lack of experimental design description: Provide more information about the experimental design, such as the number of replicates and the number of independent experiments conducted for each experiment.
2. Inconsistent use of protein expression analysis: Clarify whether protein expression levels were determined by Western blotting or immunohistochemistry.

Additional comments

1. Specify the objectives of the study more clearly and precisely, such as investigating the effect of naringenin on specific cellular processes or exploring its potential as a therapeutic intervention for osteoarthritis.

Reviewer 2 ·

Basic reporting

The manuscript exhibits a good standard of English expression, including a detailed account of technical procedures. The introduction provides a substantial overview of chondrocytes and their importance in cartilage health. However, the general context of the research in relation to existing studies on osteoarthritis is briefly touched upon and could be further expanded for better clarity. The authors adequately highlight the knowledge gap concerning the mechanism of action of naringenin and its possible effects on the TLR4/TRAF6/NF-κB pathway. The manuscript then efficiently bridges this gap through a series of comprehensive experiments. Technically, the study is well-executed, utilizing both in vitro and in vivo models. The methodology is well-described, thus facilitating future replication of the experiments. The data presented in the manuscript is robust, featuring comprehensive control measures, although deeper elaboration on the statistical measures employed would be advantageous. In conclusion, this study successfully probes the beneficial effects of naringenin on osteoarthritis, expanding on the existing knowledge and facilitating further research in this avenue.

Experimental design

1) Provide more details about the isolation process of primary chondrocytes.
2) Clarify the duration of the cell culture period.
3) Specify the volumes and concentrations of Trizol and PrimeScript RT kits used.
4) Clarify the purpose and procedure of FITC staining.
5) Clarify the meaning of "mean ± SD".

Validity of the findings

1) Insufficient information about treatment duration: Specify the duration of naringenin treatment for all experiments to ensure consistency.

Additional comments

1) Provide a brief background on previous research efforts in the field of osteoarthritis treatment using flavonoids, highlighting key findings and any existing knowledge gaps.
2) Explain the link between chondrocyte apoptosis and the pathogenesis of osteoarthritis more thoroughly, including how it contributes to the degeneration of articular cartilage.

Reviewer 3 ·

Basic reporting

The manuscript presents a well-structured discussion of the research contents; however, the complexity of the language and technical jargon used, at times, interrupted the flow and may be better suited for an audience well-versed in the intricacies of osteoarthritis research. The introduction, while providing a solid foundation on chondrocytes, their function, and negative effects of IL-1β, could have been more comprehensive in establishing the broader context of the field and defining the relevance of the study. The identification of the knowledge gap, specifically the influence of naringenin on the TLR4/TRAF6/NF-κB pathway, was well addressed. However, the progression of the study in filling that gap was not entirely convincing, in light of few shortcomings in the technical standard and methodology. The detail level of the experimental methods was somewhat lacking, potentially hindering the replication of the study. Moreover, the data presented was far from satisfactory in terms of its robustness and statistical validity. The utilized statistical methods, if any, were not explicitly mentioned, hence raising doubts about the data's credibility. I believe the data's integrity could benefit from more stringent control measures. To sum up, while the study undeniably holds potential in broadening the understanding of naringenin's impact on osteoarthritis, it falls short in crucial aspects which curtail the confidence in its findings.

Experimental design

I. Clarify the significance of the IL-1β-treated primary chondrocytes model in studying the effects of naringenin, explaining how this model mimics the inflammatory conditions observed in osteoarthritis.
II. Define the abbreviations DMEM and CO2 on their first use. Provide the brand and catalog numbers of the penicillin and streptomycin used.
III. Provide more details about the surgical procedure performed on the mice.
IV. Specify the exposure conditions for the chemiluminescence imaging system.
V. Specify the duration and temperature of the centrifugation steps.
VI. Clarify the duration of staining with hematoxylin and eosin.
VII. Explain the reasons for choosing nonparametric tests and specify their names. Consider mentioning any corrections made for multiple comparisons.

Validity of the findings

I. Specify the number of replicates performed for each experiment.
II. Insufficient details on primary chondrocyte origin: Specify the source and characteristics of the primary chondrocytes used in the study (e.g., species, tissue source, passage number).
III. Unclear statistical analysis: Describe the specific statistical tests used for data analysis, including the rationale for choosing each test.
IV. Incomplete description of caspase-3 analysis: Provide details on the method used to evaluate cleaved caspase-3 expression, such as immunohistochemistry or Western blotting, along with antibody information.

Additional comments

I. Connect the anti-inflammatory effect of naringenin to its potential therapeutic effect on osteoarthritis more clearly, explaining how reducing inflammation can impact disease progression.
II. Enhance the overall clarity of the writing by rephrasing sentences that are unclear or require more precise wording, ensuring that the scientific terminology is accurate and easily understandable.

---

## Round 0.2 · Minor Revisions

Some issues needed to be addressed:

1. Line 265: [I IL-1β induces the production] should be [IL-1β induces the production].

2. Figure 6A: The scale bar was not seen, except that 50 microns was labeled.

3. Molecular weight should be displayed in all WB results.

4. The Discussion Section remains unsatisfactory. Several paragraphs (such as line 271) lack a proper summary. The authors should give the readers a clear impression of at least how this study relates to previous literature, how the findings of this study add new knowledge, and what are the future topics that have not been adequately addressed in this study?

5. Line 263: [Apoptosis in cartilage cells does not occur simultaneously, which is consistent with the slow process of cartilage degeneration in osteoarthritis]: This sentence should be rewritten to make it easier for the readers to understand.

Reviewer 1 ·

Basic reporting

I have reviewed the author's revisions to the article and have made point by point revisions and responses according to my review comments. I believe that the article has met the publishing standards and can be published.

Experimental design

The author has made detailed modifications to the experimental design section, which is very good.

Validity of the findings

The author has made further revisions to the results section of the article, and I have no further review comments.

Additional comments

I have no further additional comments.

Reviewer 2 ·

Basic reporting

After the author's revisions, the language description of the article is clear and clear, and professional English is used throughout, and have professional article structure, charts, and tables.

Experimental design

After the author's revisions, the research questions in the article are clearly defined, relevant, and meaningful; the described method has sufficient details and information for replication.

Validity of the findings

The conclusion section of the article is fully stated and relevant to the original research question.

Additional comments

The author has made good revisions and I agree to publish this article.

Reviewer 3 ·

Basic reporting

The authors have addressed all of my comments and have incorporated those in the revised manuscript. Hence, I recommend to accept it for publication.

Experimental design

The author has made reasonable modifications to the experimental content, which is good.

Validity of the findings

The content of the results section has also been properly modified, and I believe there are no other issues.

Additional comments

Your revised manuscript is acceptable for publication.

---

## Round 0.3 · accepted · Accept

My concerns have been adequately addressed and I think this revised article can be considered for publication in this journal.